# Genomic Profiling for Predictive Treatment Strategies in Fibrotic Interstitial Lung Disease

**DOI:** 10.3390/biomedicines12071384

**Published:** 2024-06-21

**Authors:** Fabio Perrotta, Stefano Sanduzzi Zamparelli, Vito D’Agnano, Antonia Montella, Ramona Fomez, Raffaella Pagliaro, Angela Schiattarella, Mario Cazzola, Andrea Bianco, Domenica Francesca Mariniello

**Affiliations:** 1Department of Translational Medical Sciences, University of Campania “L. Vanvitelli”, 80131 Naples, Italy; vito.dagnano94@gmail.com (V.D.); antonia.montella@studenti.unicampania.it (A.M.); ramona.fomez@studenti.unicampania.it (R.F.); raffaella.pagliaro@studenti.unicampania.it (R.P.); angela.schiattarella@studenti.unicampania.it (A.S.); andrea.bianco@unicampania.it (A.B.); 2Unit of Respiratory Medicine “L. Vanvitelli”, A.O. dei Colli, Monaldi Hospital, 80131 Naples, Italy; 3Division of Pneumology, A. Cardarelli Hospital, 80128 Naples, Italy; stefanosanduzzi@gmail.com; 4Unit of Respiratory Medicine, Department of Experimental Medicine, University of Rome “Tor Vergata”, 00133 Rome, Italy; mario.cazzola@uniroma2.it; 5UOC Pneumotisiologia Federico II, A.O.R.N. Monaldi-Cotugno-CTO Piazzale Ettore Ruggieri, 80131 Napoli, Italy; nikamariniello93@gmail.com

**Keywords:** pharmacogenomics, interstitial lung diseases, idiopathic pulmonary fibrosis, antifibrotics, nintedanib, pirfenidone, precision medicine

## Abstract

Idiopathic pulmonary fibrosis (IPF) has traditionally been considered the archetype of progressive fibrotic interstitial lung diseases (f-ILDs), but several other f-ILDs can also manifest a progressive phenotype. Integrating genomic signatures into clinical practice for f-ILD patients may help to identify patients predisposed to a progressive phenotype. In addition to the risk of progressive pulmonary fibrosis, there is a growing body of literature examining how pharmacogenomics influences treatment response, particularly regarding the efficacy and safety profiles of antifibrotic and immunomodulatory agents. In this narrative review, we discuss current studies in IPF and other forms of pulmonary fibrosis, including systemic autoimmune disorders associated ILDs, sarcoidosis and hypersensitivity pneumonitis. We also provide insights into the future direction of research in this complex field.

## 1. Introduction

Interstitial lung disease (ILD) is a collective term that covers over 200 different diffuse lung diseases characterised by varying degrees of inflammation and fibrosis of the lung parenchyma, each with different clinical courses, treatments, and prognoses. Despite the rarity of individual ILDs, they collectively represent a global burden, with overall incidence and prevalence rates ranging from 9.4 to 83.6 per 100,000 person-years and 33.6 to 247.4 per 100,000 people, respectively [1]. The symptoms and impaired health-related quality of life associated with ILD, regardless of the underlying pathogenetic mechanisms, significantly reduce the survival expectancy of patients [2]. Idiopathic pulmonary fibrosis (IPF) is the most studied and common form of idiopathic interstitial pneumonia (IIP) and serves as the archetype of progressive, self-sustaining fibrosing lung disease. IPF, a chronic fibrosing ILD of unknown origin, is characterised by progressive scarring and architectural distortion of the lung interstitium, identified by the histopathological and/or radiological pattern of usual interstitial pneumonia (UIP) [3].

The onset of fibrosis is thought to be perpetuated by an abnormal wound healing mechanism and excessive collagen deposition by myofibroblasts triggered by repeated micro-injuries to the lung epithelium. In addition, the scientific literature has highlighted the impact of individual genetic backgrounds, such as surfactant protein-related genes, telomerase complex genes, *MUC5B*, TOLL-interacting protein (*TOLLIP*), matrix metalloproteinases (*MMPs*), integrin αvβ6 (*ITGAV*) polymorphism and microRNA (miRNA) expression levels, in influencing disease pathogenesis and clinical course [4,5,6,7,8,9,10].

Similarly, several other ILDs can progress to a progressive fibrosing phenotype despite initial treatment, leading to deterioration in lung function and gas exchange. These include idiopathic non-specific interstitial pneumonia (iNSIP), fibrosing hypersensitivity pneumonia (f-HP), fibrosing sarcoidosis and connective tissue diseases (CTD), such as systemic sclerosis-associated ILD (SSc-ILD) and rheumatoid arthritis-associated ILD (RA-ILD), as well as unclassifiable IIPs, occupational lung disease, drug-induced lung disease and exposure-related lung disease [11,12].

In non-IPF f-ILDs, an estimated 13–40% of patients have a progressive fibrotic phenotype [13]. Despite significant heterogeneity in aetiology, fibrotic ILDs (f-ILDs) share common pathways involving an exaggerated inflammatory response that triggers alterations in type II alveolar homeostasis, perpetuating downstream epithelial-mesenchymal transition signalling, ultimately leading to extracellular matrix (ECM) production and deposition and lung fibrosis [14]. Although the clinical course varies, patients with f-ILD typically experience a gradual decline in respiratory symptoms and disease progression on imaging, particularly those with UIP patterns such as RA-ILD or f-HP [15,16]. Early diagnosis and prompt, effective therapies are therefore essential to prevent clinical deterioration, functional disability, and adverse events (AEs).

Since their approval, antifibrotic agents, nintedanib and pirfenidone, have remained the cornerstone of pharmacological treatment of IPF, capable of slowing lung function decline and limiting disease progression [17,18]. Nintedanib, a tyrosine kinase inhibitor that binds to the ATP domain of profibrotic receptors, has been approved as an add-on therapy for the treatment of SSc-ILD and progressive f-ILD based on the results of two randomised clinical trials (RCTs) [19,20,21]. Conversely, the current level of evidence for pirfenidone, an anti-inflammatory and antifibrotic agent that mainly inhibits TGF-β, is insufficient to justify its use beyond IPF [22]. In other f-ILDs, which are primarily driven by inflammation, immunomodulators, such as glucocorticoids, azathioprine, mycophenolate mofetil or tacrolimus, are currently used, although in some diseases there is a lack of strong evidence to support their use in guidelines or international recommendations [19,20]. Despite the expansion of novel and combination therapies in the pharmacological landscape of f-ILDs, the response to treatment remains remarkably variable and unpredictable.

The increasing interest in precision medicine, a medical approach based on patient-tailored healthcare through the analysis of behavioural, environmental, epigenetic, and genetic factors, has led to a trend towards identifying genetic predictors of prognosis and therapeutic response in f-ILD, thereby enabling patient stratification [23].

According to the National Institutes of Health’s Precision Medicine Initiative Working Group, precision medicine is an advanced approach to medicine [24]. It aims to improve the effectiveness of disease treatment and prevention by considering individual differences in genes, environment and lifestyle. This approach is redefining our understanding of disease development and progression, leading to improved diagnosis, prevention strategies, treatment selection and innovative therapies [24]. Precision medicine aims to improve treatment effectiveness, reduce unnecessary therapies and optimise healthcare resources by identifying diagnostic, prognostic and therapeutic biomarkers for patient stratification and personalised therapy. Although personalised medicine has already demonstrated success in areas such as lung oncology, chronic obstructive pulmonary disease and asthma within respiratory medicine, its specific role in f-ILDs remains to be fully established [25,26,27].

The relatively low prevalence estimates as individual nosological entities hinder the development of sufficient evidence in this area of research. Nevertheless, recent technological advances over the last two decades have significantly influenced the study of human genetics, making it a critical component of precision medicine. One of the most recent genetic applications in this area is pharmacogenomics (PGx), which involves tailoring drug selection and dosage to the genetic characteristics of the patient [28].

PGx, defined as the study of variations in DNA and RNA characteristics in relation to drug response, focuses on exploring how variations across the entire genome contribute to individual responses to drugs. This provides mechanistic insights for the development of new therapies that improve drug efficacy while minimising AEs. It is important to distinguish PGx from pharmacogenetics (PGt) (Table 1), which is primarily concerned with inter-individual variations in DNA sequence related to drug absorption, distribution, metabolism, and excretion (pharmacokinetics) or drug action (pharmacodynamics). PGt aims to determine how polymorphic genetic variations affect susceptibility to a particular treatment [29].

To minimise the risk of misdiagnosis and inappropriate treatment, the precision medicine approach uses genetic variations encoding transporters, drug metabolism enzymes, receptors and other proteins as commonly used biomarkers for diagnostic, prognostic, and therapeutic purposes. While single nucleotide substitution (SNP) remains the most used pharmacogenetic polymorphism as a biomarker, involving the alteration of a single nucleotide in the gene sequence, other structural variations, such as insertions, deletions, copy number variations and inversions, have also been investigated. These polymorphisms are considered to be potential predictive biomarkers of drug efficacy and safety, allowing optimised, tailored treatment for individual patients [30].

In this review, we provide an overview of the current literature focusing on the specific impact of PGx in fine-tuning the regulation of immunomodulatory and antifibrotic agents in IPF and other f-ILDs.

## 2. Moving Pharmacogenomics from Bench to Bedside

PGx has allowed the identification of predictive biomarkers of drug pharmacokinetics (PKs) and pharmacodynamics (PDs) and current knowledge of genome-disease and genome-drug interactions offers the opportunity to optimize tailored drug therapy. High-throughput PGx genotyping, from targeted to more comprehensive strategies, allows the identification of PK/PD genotypes to be developed as clinical predictive biomarkers [30]. The response to drugs is highly variable among individuals. Indeed, genetic variants are estimated to affect between 20 and 95% of the response variability, depending on the drug [31]. Germline genetic variants can influence drug adsorption, distribution, metabolism, and elimination (ADME), and they can be responsible for reduced drug efficacy or increased toxicity. Patients might benefit from using PGx to inform treatment decisions regarding drug selection and dosing. The PGx approach has the potential of improving drug efficacy and/or avoiding unwanted side effects; these improvements could lead to better treatment adherence and outcomes and limit non-negligible healthcare system costs [28,32]. During recent years several national and international projects have been built to tackle these issues. Originally, the Clinical Pharmacogenetics Implementation Consortium (CPIC) and the Dutch Pharmacogenetics Working Group (DPWG) have developed validated guidelines for several drug-gene interactions, which are made freely available as an online resource (www.pharmgkb.org). Similarly, in March 2018, an Italian consortium was set up to create FARMAPRICE, a clinical decision support system (CDSS) designed to be used in the clinical setting to facilitate the use of PGx in the drug prescription process in Italy [30]. Actually, PGx guidelines for drug dosing have now become available for a wide range of medications associated with gene–drug interactions that could potentially be clinically actionable [33]. Although only a limited number of predictive biomarkers are of high priority for dose adjustment, PGx discovery and validation of predictive biomarkers represent one of the major challenges for disease management, drug development, patient outcome, and the reduction of healthcare costs from a precision medicine perspective. In this complex scenario, further efforts are needed to overcome barriers to PGx implementation, and demonstrating the utility and cost-effectiveness of PGx testing for clinicians will enable translation into clinical practice [30,34].

## 3. Idiopathic Pulmonary Fibrosis

IPF is a progressive and fatal lung disease of unknown origin. It has a poor prognosis, with a survival rate of 3–5 years from symptom onset if left untreated [35]. Evidence from familial cases of IPF suggests that genetic factors affecting mucin regulation, telomere length, surfactant proteins, oxidative stress and ECM homeostasis may influence susceptibility and key clinical outcomes of the disease [36]. The course of IPF is variable and unpredictable. There is currently no cure for IPF patients other than lung transplantation, although two different antifibrotic drugs, nintedanib and pirfenidone, have been approved to slow functional decline and progression to respiratory failure [17,18]. Both drugs can induce important AEs and are administered with little consideration of disease severity and genetic variation between patients.

### 3.1. Efficacy of Antifibrotic Agents

Since the introduction of antifibrotic treatments into clinical practice, which have transformed our understanding of IPF, some patients have remained stable for many years, while others have experienced steady progression, and a minority have experienced rapid disease worsening. Consequently, in recent years, various biomarkers associated with the intricate cellular, biological and immunobiological pathways have been studied as potential indicators of treatment efficacy [37,38,39].

Oxidative stress signalling plays a crucial role in the pathogenesis of pulmonary fibrosis. As a result, a combination of high-dose N-acetylcysteine (NAC) and immunomodulatory agents, such as oral glucocorticoids (GCs) and azathioprine, was considered the standard of care for IPF patients until 2012. However, in that year, data from the PANTHER RCT showed no significant benefit in terms of change in forced vital capacity (FVC) [37].

Subsequently, a retrospective analysis of participants in the PANTHER trial investigated whether SNPs within *TOLLIP* and *MUC5B* might influence response to NAC therapy [38]. The authors found that individuals carrying a specific genetic variant, the *TOLLIP* rs3750920 TT genotype, but not *MUC5B*, were most likely to benefit from NAC therapy. Specifically, NAC treatment was associated with a significant reduction in the risk of a composite endpoint (defined as death, transplantation, hospitalisation or ≥10% decline in FVC) in patients with the TT genotype, while a nonsignificant increase in endpoint risk was observed in patients with the CC genotype. Overall, given that this genetic variant is present in 25% of IPF patients, the therapeutic potential of NAC remains uncertain [40].

The promoter region of the *MUC5B* gene encodes a highly glycosylated mucin-5B precursor protein, which plays a role in airway rheology, mucociliary transport and immune defence in the airways [41]. The presence of the T allele in *MUC5B* rs35705950 increases the risk of developing interstitial lung abnormalities by 2.8-fold and the risk of having definite CT evidence of pulmonary fibrosis by 6.3-fold [42]. Recent studies have, therefore, sought to unravel this complex relationship.

A single-centre retrospective study evaluated the potential association of *MUC5B* and *TOLLIP* SNPs with clinical outcomes in 62 IPF patients undergoing antifibrotic treatment. No association between *MUC5B* variants and survival, disease progression or acute exacerbations was found [43]. However, it was observed that patients carrying the *TOLLIP* rs5743890 C/T genotype had a higher risk of death (*p* = 0.014) and disease progression (*p* = 0.001) compared to those with the T/T genotype.

In contrast, another research in IPF patients treated with pirfenidone showed different results [44]. This study showed that individuals with the *TOLLIP* rs5743890 CC or CT genotype had prolonged survival. Furthermore, authors documented a protective role associated with the *TGF-B1* rs1800470 GA genotype, while it identified a detrimental effect associated with the telomerase reverse transcriptase (*TERT*) rs2736100 CA genotype on overall survival. Importantly, no significant correlations were found between *MUC5B* promoter SNPs or *CYP1A2* SNPs and survival in this study.

The common polymorphism within the *MUC5B* promoter (rs35705950) was also associated with improved survival in IPF patients in the INSPIRE trial. In this trial, patients were randomly assigned to receive either interferon (IFN) gamma-1β or placebo, but the trial showed no benefit for patients treated with IFN gamma-1β [45].

In a study investigating common profibrotic variants of *MUC5B* and desmoplakin (*DSP*), IPF patients receiving antifibrotic therapies and carrying the *DSP* rs2076295 G allele or the *MUC5B* rs35705950 T allele had reduced mortality [46]. DSP serves as a constituent of desmosomes, which are biological structures involved in cell–cell adhesion that also modulate cell proliferation, differentiation, migration, and apoptosis. Variants of *DSP* rs2076295 contribute to IPF pathogenesis by inducing overexpression of *DSP*, while *DSP* rs2076295 G alleles are associated with decreased gene expression in IPF lung tissue compared to the TT genotype [47]. Interestingly, the G allele in *DSP* rs2076295 was discovered to confer a protective effect on mortality and lung function decline in the IPF population treated with nintedanib, whereas the TT genotype for the same gene conferred an overall survival benefit in patients treated with pirfenidone [46]. No differences in lung function decline or overall survival were observed in IPF patients with the *MUC5B* rs35705950 genotype based on the different antifibrotic drugs.

These results regarding the *MUC5B* promoter and the rs35705950 genotype were further supported by another single-centre study, which showed a prolonged survival in IPF patients with the T allele (TT or TG versus GG) [48].

### 3.2. Safety of Antifibrotic Agents

Telomere length in IPF patients has been extensively studied, primarily because of its robust correlation with disease risk and severity. The PANTHER-IPF clinical trial, which was prematurely stopped due to evidence of increased mortality, hospitalisations, and serious treatment-related AEs, revealed a notable association between immunosuppression and outcomes in individuals categorised by leukocyte telomere length (LTL) [49]. Specifically, the study showed that exposure to immunosuppressants was associated with an increased composite endpoint of death, lung transplantation, hospitalisation or decline in FVC in patients with an LTL below the 10th percentile.

Telomere-mediated diseases typically present in a syndromic pattern, manifesting with extrapulmonary symptoms such as bone marrow failure, liver cirrhosis, hearing loss or skin and haematological malignancies [50,51]. Due to the relatively young age of onset of IPF, patients with telomerase mutations are often considered candidates for lung transplantation [52]. However, individuals with telomerase deficiency who undergo lung transplantation often experience rare and severe systemic complications leading to unfavourable outcomes. In addition, they have poor tolerance to long-term immunosuppressive drug regimens due to the additional myelosuppressive, hepatotoxic and carcinogenic effects required to prevent allograft rejection [53,54].

Two small cohort studies of 8 and 9 patients investigated the clinical outcomes of lung transplantation in individuals with pulmonary fibrosis carrying mutated telomerase genes [55,56]. Notably, both studies observed a notable incidence of haematological complications, particularly thrombocytopenia, requiring blood transfusions and adjustments in immunosuppressive doses. In addition, Silhan et al. reported a high prevalence of acute renal failure related to tubular injury and calcineurin inhibitor nephrotoxicity, while 37% of patients experienced gastrointestinal bleeding attributed to mycophenolate-induced ischaemic colitis [56]. Borie et al. documented a significant reduction in the median survival of their patients (214 days) compared to the survival rate reported by the International Society for Heart and Lung Transplantation in lung transplant recipients with pulmonary fibrosis, who typically have a median survival of 4.3 years [55,57]. The mammalian target of rapamycin inhibitors may provide a less nephrotoxic alternative to calcineurin inhibitors, although their use may be limited by myelosuppressive effects [58].

Azathioprine is another cytotoxic drug given to lung transplant recipients, with reported cases of fulminant hepatitis associated with its use in patients with telomere defects [59]. Tokman et al. substantiated these findings by describing the clinical outcomes of 14 lung transplant recipients with telomerase mutations. Following lung transplantation, almost 80% of patients developed bone marrow failure and renal dysfunction was more common than expected. However, it is worth noting that both haematological and renal abnormalities were mild in many subjects [60].

A multicentre study of a cohort of 38 lung transplant patients with telomerase-related gene (*TEG*) mutations suggests that haematological risk is higher in individuals with *TERT* or *TERC* mutations [61]. The clinical significance of the remarkable role of the *TERT* gene was highlighted by a case report of a very rare mutation (*TERT 3100C > T*) identified in a woman who underwent lung transplantation for familial pulmonary fibrosis and subsequently developed acute graft-versus-host disease [62]. This mutation appears to result in impaired immune surveillance and defective cellular immunity, thereby reducing the ability to eliminate donor immune cells. Furthermore, patients with IPF and telomere dysfunction are at increased risk of opportunistic infections due to impaired in vitro *cytomegalovirus*-specific T-cell effector function [52].

DNA samples from patients enrolled in the Anti-Coagulant Effectiveness in Idiopathic Pulmonary Fibrosis trial were used in a genome-wide association study [63]. Examination of this group revealed a possible association between warfarin treatment and the *MUC5B* promoter polymorphism, although the sample size was insufficient for formal interaction testing [39]. Notably, among the genotyped study participants, a greater number of deaths were observed in individuals with the polymorphism who received warfarin compared to those who received placebo (6 on warfarin vs. 1 on placebo), suggesting a possible interaction.

Finally, relaxin is a hormone with the potential to degrade collagen fibres. Tan et al. conducted an analysis of relaxin/insulin-like family peptide receptor 1 (*RXFP1*) gene expression in lungs affected by idiopathic pulmonary fibrosis (IPF) and control lungs. The researchers found a significant reduction in *RXFP1* gene expression in IPF cases. They also showed that CGEN25009, a relaxin-like peptide, requires *RXFP1* expression. Therefore, only a subset of patients with high RXFP1 expression would benefit from therapies involving this peptide. In addition, the progression of IPF may be associated with a gradual decline in *RXFP1* expression, suggesting the potential benefit of treatment with relaxin agonists early in the course of the disease [64]. Figure 1 summarise the potential influence of genetic loci modulating the response to treatment in patients with IPF harbouring different allele variants.

## 4. Connective Tissue Disease Related Interstitial Lung Disease

CTD-related interstitial lung disease (CTD-ILD) is a major cause of morbidity and mortality in CTD. Although the full manuscript is not yet available, a recent update of the American College of Rheumatology guideline for the management of ILD associated with systemic autoimmune rheumatic diseases, including SSc, RA, idiopathic inflammatory myopathies, mixed connective tissue diseases, and primary Sjögren’s syndrome (pSS), has been published [65]. This guideline summary provides evidence-based recommendations for primary ILD therapies and for progressive ILD. Specifically, for people with SSc-ILD, options, such as mycophenolate mofetil, tocilizumab, a humanized anti-IL-6 receptor monoclonal antibody (mAb), or rituximab, a mAb that targets the B-cell marker CD20 and leads to B-cell depletion, are recommended as primary choices, with a strong recommendation against the use of GCs. Conversely, mycophenolate mofetil, azathioprine and rituximab emerge as potential primary therapies for RA-ILD, mixed CTD-ILD and pSS-associated ILD, where a short course of GCs may be incorporated. Additional second-line treatments include cyclophosphamide and JAK inhibitors. Patients with CTD-ILD who have the PF-ILD phenotype and those with SSc-ILD who continue to progress despite initial therapy may benefit from the addition of nintedanib [19,20].

The AEs of these drugs, the dosage of treatment and the clinical course can vary significantly between patients, highlighting the urgent need for individualisation of therapy. A summary of the evidence presented below is shown in Table 2.

### 4.1. Efficacy

GCs are often used as an adjunct to other immunomodulatory therapies in people with CTD-ILD. The efficacy of GC treatment may be influenced by polymorphisms in P-glycoprotein (*P-gp*) genes, key enzymes involved in transmembrane transport [83]. In particular, certain polymorphisms (3435C > T, 2677G > T and 1236C > T) in the *P-gp* gene have been associated with impaired transport activity in RA patients, thereby reducing intracellular drug concentrations [66,67]. P-gp is also involved in the transport of methotrexate, a cornerstone of RA treatment, and the same 3435C > T polymorphism in exon 26 has been associated with response to this agent [68].

Several interesting studies have investigated predictors of response to rituximab. In one study, low baseline expression of IFN type I response genes (*IRGs*) (*LY6E*, *HERC5*, *IFI44L*, *ISG15*, *MxA*, *MxB*, *EPSTI1* and *RSAD2*) was associated with a favourable clinical response to rituximab [69]. IFNs type I play a critical role in immunological processes such as lymphoid differentiation, homeostasis, tolerance, and memory. Response to rituximab may also be influenced by polymorphisms in the Fc gamma Receptor (*FCGR*) gene, which encodes FcγRIIIA. In particular, the 158V/F polymorphism of FcγRIIIA, which substitutes a valine (V) for a phenylalanine (F) at amino acid position 158, was independently associated with a good clinical response to rituximab in RA patients [70]. However, this polymorphism did not predict the efficacy of rituximab in patients with pSS [77]. The homozygous F/F phenotype has been associated with a more favourable response to infliximab, adalimumab and etanercept in RA [71].

Further investigation has shown that the B-cell activating factor (*BAFF*)-871C > T polymorphism influences the response to rituximab in RA patients, with the CC genotype being significantly associated with a higher response rate [70]. Polymorphisms in the promoter region of the B lymphocyte stimulator (*BLyS*) gene have also been investigated as potential markers of response to rituximab in RA patients, with the TTTT haplotype identified as a genetic marker associated with a good response to rituximab in patients seropositive for rheumatoid factor and/or anti-cyclic citrullinated peptide [72].

IL-6 plays a key role in B-cell proliferation and joint destruction in RA, and a significant correlation between the -174G > C IL-6 promoter polymorphism and response to rituximab in RA patients has been demonstrated [73]. Specifically, patients carrying the CC genotype had a significantly poorer response to rituximab than patients carrying the GC/GG genotype. Interestingly, serum IL-6 levels did not correlate with the different IL-6 genotypes, suggesting that IL-6 at baseline cannot predict response to rituximab.

Tocilizumab has been the subject of limited studies investigating the role of genetic variation in relation to treatment response. In particular, the expression of IFN type I genes appears to influence response to tocilizumab, with increased expression of three genes (*IFI6*, *MX2* and *OASL*) associated with a favourable clinical response to TCZ [74] [Sanayama et al., 2014]. In a cohort of 13 RA patients treated with TCZ, changes in the expression of four genes (*CCDC32*, *DHFR*, *EPHA4* and *TRAV8-3*) at baseline were found to determine responder status [84].

Mycophenolate mofetil is rapidly converted to mycophenolic acid (MPA), a selective inhibitor of inosine monophosphate dehydrogenase (IMPDH) expressed by activated lymphocytes [75]. The presence of the rs11706052 SNP in intron 7 of the *IMPDH2* gene has been shown to significantly reduce the efficacy of MPA in inhibiting lymphocyte proliferation in vitro. Further investigation is warranted to elucidate the implications of these findings.

### 4.2. Safety

Azathioprine and mycophenolate mofetil, which block B and T lymphocyte production and proliferation, respectively, are commonly used in the treatment of CTD-ILD. Despite its efficacy, azathioprine has significant AEs, including gastrointestinal intolerance, myelotoxicity, pancreatitis, and hepatic dysfunction. Azathioprine is an inactive prodrug that is rapidly converted to active 6-mercaptopurine (6-MP). The enzyme thiopurine methyltransferase (TPMT) inactivates 6-MP, reducing its cytotoxic effects. *TPMT* polymorphisms may increase the risk of AEs, with low TPMT activity being associated with bone marrow toxicity and gastric intolerance, but not hepatotoxicity [85]. Three allelic variants of *TPMT*, *TPMT2*, *TPMT3A* and *TPMT3C*, account for 80–90% of individuals with low or intermediate TPMT activity [86,87]. International clinical recommendations and guidelines differ on the determination of TPMT status prior to initiation of azathioprine therapy, although regular monitoring of white blood cells is recommended as AEs may be TMPT-independent [39,88]. Patients with RA who are heterozygous for the *TPMT3A* allele have an increased risk of azathioprine-related haematopoietic and gastrointestinal toxicity compared to those with the wild-type allele [89,90]. A recent study investigated whether genetic variants in genes encoding enzymes involved in the methotrexate and azathioprine metabolic pathways, such as *TPMT*, *MTHFR* and *SLCO1B1*, are associated with disease severity in SSc patients receiving immunosuppressive drugs [76]. The authors confirmed the hypothesis that carriers of the alternative *TPMT*3A* allele have reduced TPMT enzyme activity, resulting in toxic metabolites and consequent AEs when azathioprine is administered. In addition, a missense C677T variant in the *MTHFR* gene, rs1801133, has been associated with the risk of high systolic blood pressure in patients receiving azathioprine therapy. Rs1801133 may also serve as a predictor of response to GCs, as the efficacy of GCs appears to be increased when this variant is present in children with asthma [91].

Taha et al. conducted a comparison of the rate of azathioprine discontinuation due to serious AEs in a cohort of patients with ILD who were genotyped (*TPMT* genotyping) and an untested cohort [88]. The authors found that the overall incidence of AEs was similar between the two cohorts, but the discontinuation rate was significantly lower in the genotyped cohort compared to the untested cohort. Thirty-five patients in the genotyped cohort (71%) were also tested for human leukocyte antigen (*HLA)-DQa1-HLA-DRB1* because an association between the HLA gene region and azathioprine-induced pancreatitis has been identified in patients with inflammatory bowel disease [92]. Specifically, a significant increase in azathioprine-induced pancreatitis in individuals with inflammatory bowel disease who carry the A/C or C/C variant alleles for the SNP rs2647087 of *HLA* class II *DQA1* was demonstrated compared to those who are homozygous for the common allele (A/A).

Short leukocyte telomere length (LTL) has been reported in patients with several ILDs and correlates with reduced survival and rapid decline in lung function [93,94]. A multicentre cohort analysis investigated the potential pharmacogenomic interaction between LTL and immunosuppressants (mycophenolate mofetil or azathioprine) in patients with f-HP, unclassifiable ILD (uILD) and CTD-ILD [95]. However, as only 12% of CTD-ILD patients in this study had LTL below the 10th percentile, it was not possible to investigate the survival associations between the use of immunosuppressants and LTL in CTD-ILD patients.

In patients with SSc-ILD, ACR guidelines strongly discourage the use of GCs. In particular, high doses of GCs (>15 mg daily of prednisone or equivalent) have been associated with the development of scleroderma renal crisis, a life-threatening complication in people with SSc [96]. The incidence of scleroderma renal crisis tends to be higher in patients with positive anti-RNA polymerase III autoantibodies (ARAs) [97]. In a large cohort of patients with SSc, the predictive role of HLA genetic markers for scleroderma renal crisis was investigated. *HLA-DRB10407* and *HLA-DRB11304* were identified as independent risk factors for the development of scleroderma renal crisis, but data on the use of GCs in these patients were not available [78]. Another study examined protein expression in renal biopsy specimens from ARA-positive patients with scleroderma renal crisis and found increased expression of two proteins, GPATCH2L and CTNND2, in scleroderma renal crisis compared to healthy kidney tissue. GPATCH2L protein was predominantly localised to tubular and vascular endothelial structures, while CTNND2 showed higher expression in endothelial cells. Among the nine autosomal selected SNPs in ARA-positive individuals, only one SNP (rs935332) in the *GPATCH2L* region on chromosome 14 showed an association with scleroderma renal crisis development [79]. The same gene, *CTNND2*, has also been associated with pulmonary arterial hypertension in SSc. It is conceivable that variants in the *CTNND2* gene may play a role in the dysregulation of endothelial progenitor cells observed in vasculopathic complications of SSc, such as pulmonary arterial hypertension and scleroderma renal crisis [98].

## 5. Other Fibrotic Interstitial Lung Diseases

Sarcoidosis is a systemic granulomatous disease of unknown aetiology that predominantly affects the lungs and lymphatic system, although almost all organs can be affected [99]. Treatment options for sarcoidosis vary from observation without intervention to the use of various therapeutic agents. Current guidelines recommend treatment for symptomatic disease, significant pulmonary involvement, or when critical organs, such as the eyes and heart, are affected. GC therapy remains the first-line treatment of choice. However, in cases where patients continue to experience disease progression or relapse despite glucocorticoid or other immunosuppressive treatment, the addition of TNF-α inhibitors such as infliximab or adalimumab may be considered [100].

TNF-α, a potent proinflammatory cytokine produced primarily by alveolar macrophages, plays a critical role in regulating and perpetuating granuloma formation in sarcoidosis. Elevated levels of TNF-α often correlate with disease activity and progression [101,102]. However, the use of TNF-α inhibitors in the treatment of sarcoidosis is challenging due to potential side effects and the significant costs associated with treatment. Therefore, there is increasing interest in identifying patients who are more likely to develop severe disease and would benefit from biological treatment targeting TNF-α.

Previous studies have suggested that polymorphisms in genes responsible for TNF-α production may have prognostic implications for the course of sarcoidosis. In particular, the presence of the TNF-α -308A variant allele (GA/AA genotypes) has been observed in patients with Löfgren’s syndrome, a subgroup often characterised by spontaneous resolution of the disease [80]. Conversely, the GG genotype has been associated with a less favourable prognosis in sarcoidosis [81,82]. Based on this evidence, Wijnen et al. investigated the association between the presence of the *TNF-α G 308A* polymorphism and response to TNF-α inhibitors (adalimumab or infliximab) in 111 patients with refractory sarcoidosis. The authors found that patients with the *TNF-α G-308A* GG genotype, which indicates an unfavourable prognosis, had a better response to TNF inhibitors compared to carriers of the A allele (AA and GA genotypes) [103].

Sarcoidosis is primarily characterised as a T-cell-mediated disease involving the production of inflammatory cytokines; however, there is evidence that humoral immunity may also play a role in its pathogenesis [104]. In a phase I/II clinical trial, rituximab, a monoclonal antibody targeting CD20^+^ B cells, was evaluated in patients with refractory pulmonary sarcoidosis, but clinical response to therapy was variable [105]. Further research is warranted, as suggested by findings in patients with RA that clinical response to rituximab may be related to baseline expression levels of IFN type I response genes [69].

However, a single trial investigating the effectiveness of ustekinumab, a mAb that blocks the p40 subunit of interleukin (IL)-12 and IL-23, in chronic sarcoidosis failed to demonstrate efficacy, possibly due to variations in the *IL-12* and *IL-23R* genes [106,107,108].

HP is an immune-mediated interstitial lung disease caused by the inhalation of various antigens in susceptible individuals, resulting in inflammation, granuloma formation and, in some cases, fibrosis. Treatment of HP includes avoidance of the triggering antigen, use of immunosuppressive agents and use of anti-fibrotic therapy for chronic f-HP with a progressive phenotype [109].

Similar to IPF, short LTL has been associated with a high risk of disease progression, fibrosis and poor survival in patients with HP [110]. LTL also appears to influence response to immunosuppressive therapies. F-HP patients treated with mycophenolate mofetil showed different survival rates when stratified by telomere quartiles [111]. Specifically, individuals with LTL in the first quartile were more likely to undergo lung transplantation. Furthermore, the survival of these patients showed no difference between those who received mycophenolate mofetil and those who did not (*p* = 0.87). Conversely, survival was significantly higher in patients with LTL in the second to fourth quartile who received mycophenolate mofetil compared to those in the first quartile (*p* = 0.007). The authors did not assess the potential AEs of mycophenolate mofetil treatment.

The authors also performed a retrospective, multicentre cohort analysis in f-HP, uILD, CTD-ILD patients [95]. Their research showed that f-HP and uILD patients with LTL <10th percentile experienced reduced survival when exposed to either mycophenolate mofetil or azathioprine Therefore, caution should be exercised when considering immunosuppressive treatment in these patients. In conclusion, the measurement of LTL could provide valuable information for individualising the management of patients with f-HP.

## 6. Current Ongoing Trials and Future Perspective

As previously discussed, telomere dysfunction plays a pivotal role in the development of IPF. In a phase 1–2 prospective study conducted by Townsley et al., patients with telomeropathies were administered the synthetic sex androgen hormone danazol at a dose of 800 mg daily [112]. This intervention resulted in an increase in LTL and haematological response in the participants. Of the 27 patients enrolled, 25 had clinical or subclinical pulmonary fibrosis. While the treatment was generally well tolerated, some patients required dose reductions. However, the study was stopped early after the prespecified sample size requirements were met due to the positive effects observed on telomere lengthening. These results led to the initiation of an ongoing Phase 2 trial (NCT03312400) to study the safety and effect of low-dose danazol, with the primary endpoint being the reduction of telomere attrition rate (decreased rate of telomere attrition by 50%, as compared to the baseline rate). This trial is expected to be completed by October 2027. Of relevance to IPF is the ongoing TELO-SCOPE trial (NCT04638517), a multicentre, Australian, double-blind, placebo-controlled, randomised trial in patients with pulmonary fibrosis and short telomeres. Participants will be randomised to receive either danazol or a matched placebo for 12 months, with the primary endpoint being the change in telomere length at 12 months. The estimated study completion date is June 2025.

NCT04300920 is a prototype PGx-guided RCT. This ongoing study, known as PRECISIONS (Prospective Treatment Efficacy in IPF Using Genotype for NAC Selection), focuses on IPF patients with the rs3750920 TT genotype. These patients will be randomised to receive either NAC 600 mg or placebo in addition to standard therapy. The study will compare the time to a composite endpoint of relative decline in lung function [10% relative decline in FVC, first respiratory hospitalization, lung transplantation, or all-cause mortality. Subgroup analyses based on antifibrotic use are planned. Results are expected in 2026.

The PRECISION-ILD study (NCT05998512) aims to explore the variations and similarities in genetic, genomic, and environmental factors among fibrotic ILDs, considering the specific disease type, its progression (fibrosis), and response to treatment. The goal is to incorporate the most influential biomarkers affecting prognosis and treatment response into diagnostic protocols. Additionally, the study seeks to evaluate the feasibility and cost-effectiveness of implementing a P4 (predictive, preventive, personalised, and participatory) approach in clinical practice for fibrotic ILDs, utilizing data from the Spanish SEPAR ILD Reg, Observatory IPF. cat, CIBERES IPF, and Familial ILD cohorts. Genomic information and biomarkers will be gathered to evaluate responses to various therapies.

## 7. Conclusions

PGx research is providing valuable insights for people with ILDs. Through the identification of various single nucleotide polymorphisms and specific genotypes associated with genes involved in profibrotic biological mechanisms, such as mucin regulation, telomere attrition, cell–cell adhesion and oxidative stress, studies have supported the concept of increased susceptibility and severity in IPF. Building on these discoveries, numerous investigations have sought to explore the potential of pharmacogenomics to identify ILD patients who may benefit more from antifibrotic or immunomodulatory treatments with improved safety profiles. Furthermore, upcoming research initiatives will integrate genomics into RCT patient selection to provide internal validation of treatment effects. This approach aims to refine the precision and efficacy of ILD treatments by tailoring them to individual patient profiles.

## Figures and Tables

**Figure 1 biomedicines-12-01384-f001:**
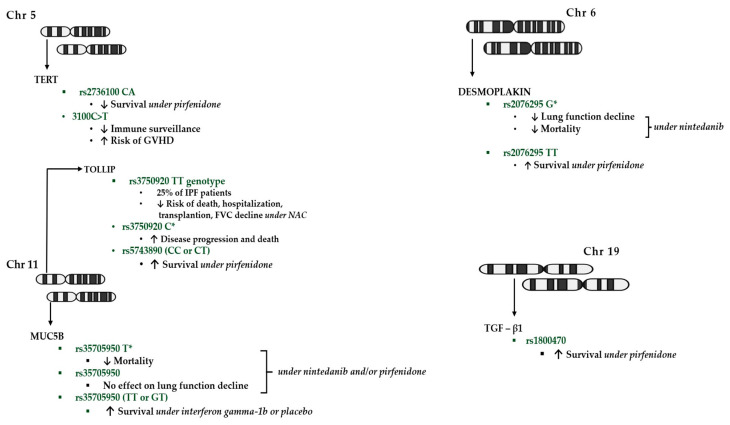
Pharmacogenomic implications of single nucleotide polymorphisms (SNPs) within the main genes involved in IPF. SNPs and specific genotypes in TERT, TOLLIP, MUC5B, Desmoplakin and TGF-β1 are associated with different responses to antifibrotics, muco-regulatory/antioxidant therapies and/or increased susceptibility to adverse events after lung transplantation. Chr: Chromosome; FVC: Forced vital capacity; GVHD: Graft-versus-host disease; IPF: Idiopathic pulmonary fibrosis; MUC5B: Mucin-5B; NAC: N-Acetylcysteine; TERT: Telomerase reverse transcriptase; TGF-β1: transforming growth factor beta 1; TOLLIP: Toll-interacting protein; ↑: increased; ↓: decreased.

**Table 1 biomedicines-12-01384-t001:** Differences in definition, focus, methodology and applications between Pharmacogenetics (PGt) and Pharmacogenomics (PGx).

	Pharmacogenetics (PGt)	Pharmacogenomics (PGx)
**Definition**	It is the study of how variations in an individual’s DNA sequence affect their response to drugs.	It is the broader study of how an individual’s entire genome, including DNA and RNA, influences their response to drugs.
**Focus**	PGt primarily examines variations in specific genes known to impact drug metabolism, transport, and receptor interactions.	PGx investigates how variations across the entire genome affect drug response, encompassing not only coding regions but also non-coding regions that regulate gene expression.
**Methodology**	It investigates single genetic variants, such as single nucleotide polymorphisms (SNPs), in key drug-metabolizing enzymes, transporters, or receptors.	It employs genome-wide approaches to analyse genetic variations, including SNPs, insertions/deletions, copy number variations, and structural variants.
**Applications**	PGt is used to predict how individuals may respond to specific drugs based on their genetic makeup, helping to tailor medication dosages and select the most appropriate treatments for patients.	PGx provides a more comprehensive understanding of the genetic basis of drug response, facilitating the discovery of novel biomarkers and therapeutic targets. It can also inform personalized medicine approaches by considering a broader range of genetic factors influencing drug efficacy and safety.

**Table 2 biomedicines-12-01384-t002:** Pharmacogenomic implications of gene variants on drug efficacy and safety in CTDs and other non-IPF-ILDs.

CTD	Gene	Chromosome	SNP Description Genetic Alteration	Clinical Consequences	Refs.
*RA*	P-gp	7q21.12	3435C > T 2677G > T 1236C > T	↓ CSs transport activity ↓ Intracellular CSs concentrations	[66,67]
	P-gp	7q21.12	3435C > T	↓ MTX response	[68]
	IRGs	/	↑ Expression	↓ RTX response	[69]
	FCGR3A	1q23.3	158V > F; FF 158 (FF)	↑ MTX response ↑ Infliximab, adalimumab and etanercept response	[70,71]
	BAFF BAFF (promoter)	13q33.3 13	871 (CC) TTTT	↑ RTX response ↑ RTX response in seropositive pts	[70,72]
	IL-6 (promoter)	7	174 (CC)	↓ RTX response compared to GC/GG	[73]
	IFI6	1p35.3	↑ Expression	↑ TCZ response	[74]
	MX2	21q22.3	↑ Expression	↑ TCZ response	[74]
	OASL	12q24.31	↑ Expression	↑ TCZ response	[74]
	IMPDH	7q32.1	rs11706052	↓ MPA efficacy to inhibit lymphocyte proliferation (in vitro)	[75]
	TMPT	6q22.3	TPMT*3A	↑ AZA-related hematopoietic and gastrointestinal toxicity	[76]
*SS*	FCGR3A	1q23.3	158V > F 158 (FF)	No effect on RTX response	[77]
*SSc*	TMPT	6q22.3	rs1801133	↑ Risk for high systolic pressure in pts under AZA	[76]
	HLA-DRB1	6q21.32	* 1304 * 0407	↑ Risk SRC	[78]
	GPATCH2L	14q24.3	rs935332	↑ Risk SRC in ARA-positive pts	[79]
Sarcoidosis	TNF-α		308 (GA/AA) 308 (GG)	↑ Probability of spontaneous resolution ↑ Risk of negative prognosis	[80,81,82]

ARA: Anti-RNA polymerase III antibodies; AZA: azathioprine; BAFF: B-Cell Activating Factor; CSs: Corticosteroids; CTDs: Connective Tissue Diseases; FCGR3A: Fc region receptor III-A; I GPATCH2L: G-Patch Domain Containing 2 Like; HLA-DRB1: Major Histocompatibility Complex, Class II, DR Beta 1; IFI6: Interferon alpha-inducible protein 6; ILDs: Interstitial lung diseases; IRGs: Interferon regulated genes; IMPDH: Inosine Monophosphate Dehydrogenase 1; MPA: Mycophenolic acid; MTX: Methotrexate; OASL: 2′-5′-Oligoadenylate Synthetase Like; P-gp: P-glycoprotein; pts: patients; RA: Rheumatoid arthritis; RTX: Rituximab; SRC: Scleroderma renal crisis; SS: Sjögren’s syndrome SSc: systemic sclerosis; TPMT: Thiopurine S-Methyltransferase; TCZ: Tocilizumab; ↑: increased; ↓: decreased.

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
