# Peer review of "Genomic Profiling for Predictive Treatment Strategies in Fibrotic Interstitial Lung Disease"

_biomedicines, 2024, doi:10.3390/biomedicines12071384_

Round 1
Reviewer 1 Report
Comments and Suggestions for Authors
In this paper, the authors presented a narrative review on genomic profiling in patients with fibrotic ILD. The paper presents the latest modern data on the relationship between genetic changes and the course of diseases, as well as the response to therapy in patients with IPF, systemic autoimmune rheumatic diseases, sarcoidosis and HP. The article is interesting, well written and clear enough, and I have no any concerns.
This is one of the most comprehensive reviews on this issue to date. Data on the safety of immunosuppressive therapy in patients with certain sets of nucleotide polymorphisms deserve special attention.
This article quite comprehensively collects all the known results of studies on the genetic basis of patients’ predisposition to the development of pulmonary fibrosis, the course of the disease and response to therapy.
All conclusions of this article are well justified based on the data presented. The authors have fully answered the main objective of this paper.
The reference includes 114 sources and very fully reflects the majority of studies devoted to the problem of the relationship between genetic changes and pulmonary fibrosis.
The tables and figures provided make this review easier to understand. It is especially necessary to note the originality of Figure 1, where all currently known data on the nucleotide polymorphism genotype in patients with IPF are presented.
Author Response
Dear Referee,
Many thanks for your evaluation of our manuscript.
In the revised version minimal changes are present including more appropriate figure legends.
We hope you may consider this version for publication.
Your Sincerely
Reviewer 2 Report
Comments and Suggestions for Authors
This is a comprehensive Review on IPF and patients predisposed to fibrosis development and predicting the progression of IPF and non-IPF ILDs. Focus on Pharmacogenomics (PGx) tailoring drug selection and dosage based on genetic characteristics
The following issues have to be considered:
1. Abstract: Please shorten to the essential message.
2. Graphical abstract overview focussed on non-IPF-ILDs would be great.
3. Overview of the current literature focusing on the specific impact of PGx in fine-tuning.
4. Table 1 Please add legend to distinguish PGx from PGt.
5. Table 2 Please add a comprehensive figure legend.
6. Figure 1 Title and expand the legend to understand the figure.
Author Response
Dear Referee,
Many thanks for your valuable review of our manuscript "Genomic Profiling for Predictive Treatment Strategies in Fibrotic Interstitial Lung Disease". We are now submitting a revised version of the manuscript based on your comments. In particular:
The following issues have to be considered:
- Abstract: Please shorten to the essential message
Many thanks for your comment. The abstract has been rephrased and shortened
- Graphical abstract overview focussed on non-IPF-ILDs would be great.
Many thanks for this comment. We have now introduced a graphical abstract about the main mechanisms influencing the response to treatment and the safety in non IPF ILDs
- Overview of the current literature focusing on the specific impact of PGx in fine-tuning
Many thanks for this comment. We have now included a paragraph about the role of the pharmacogenomics in the fine tune regulation of treatment response.
- Table 1 Please add legend to distinguish PGx from PGt.
- Table 2 Please add a comprehensive figure legend.
- Figure 1 Title and expand the legend to understand the figure
Many thanks for the above-reported comments. We have checked and changed the tables and figure legends accordingly.
Round 2
Reviewer 2 Report
Comments and Suggestions for Authors
The authors answered the question raised. The revised manuscript has gained in quality and includes a graphical abstract.